# Thrombelastography and Conventional Coagulation Markers in Chronic Obstructive Pulmonary Disease: A Prospective Paired-Measurements Study Comparing Exacerbation and Stable Phases

**DOI:** 10.3390/ijms25042051

**Published:** 2024-02-08

**Authors:** Ema Rastoder, Peter Kamstrup, Caroline Hedsund, Alexander Jordan, Pradeesh Sivapalan, Valdemar Rømer, Frederikke Falkvist, Sadaf Hamidi, Elisabeth Bendstrup, Søren Sperling, Maria Dons, Tor Biering-Sørensen, Casper Falster, Christian B. Laursen, Jørn Carlsen, Jens-Ulrik Stæhr Jensen

**Affiliations:** 1Section of Respiratory Medicine, Department of Medicine, Copenhagen University Hospital Herlev-Gentofte, 2900 Hellerup, Denmark; ema.rastoder@regionh.dk (E.R.); alexander.ryder.jordan@regionh.dk (A.J.); pradeesh.sivapalan.02@regionh.dk (P.S.); valdemar.roemer@regionh.dk (V.R.); frederikke.stenner.falkvist@regionh.dk (F.F.);; 2Department of Clinical Medicine, Faculty of Health and Medical Sciences, University of Copenhagen, 2200 Copenhagen, Denmark; joern.carlsen@regionh.dk; 3Department of Respiratory Diseases and Allergy, Aarhus University Hospital, 8200 Aarhus, Denmark; karbends@rm.dk (E.B.); sonpde@rm.dk (S.S.); 4Department of Clinical Medicine, Aarhus University, 8200 Aarhus, Denmark; 5Cardiovascular Non-Invasive Imaging Research Laboratory, Department of Cardiology, Herlev & Gentofte Hospital, University of Copenhagen, 2900 Hellerup, Denmarktor.biering@gmail.com (T.B.-S.); 6Center for Translational Cardiology and Pragmatic Randomized Trials, Department of Biomedical Sciences, Faculty of Health and Medical Sciences, University of Copenhagen, 2900 Hellerup, Denmark; 7Odense Respiratory Research Unit (ODIN), Department of Clinical Research, University of South Denmark, 5000 Odense, Denmark; casper.falster@rsyd.dk (C.F.); christian.b.laursen@rsyd.dk (C.B.L.); 8Department of Respiratory Medicine, Odense University Hospital, 5000 Odense, Denmark; 9Department of Cardiology, Rigshospitalet, Copenhagen University Hospital, 2100 Copenhagen, Denmark

**Keywords:** COPD, exacerbation, thrombelastography

## Abstract

Chronic Obstructive Pulmonary Disease (COPD) exacerbation is known for its substantial impact on morbidity and mortality among affected patients, creating a significant healthcare burden worldwide. Coagulation abnormalities have emerged as potential contributors to exacerbation pathogenesis, raising concerns about increased thrombotic events during exacerbation. The aim of this study was to explore the differences in thrombelastography (TEG) parameters and coagulation markers in COPD patients during admission with exacerbation and at a follow-up after discharge. This was a multi-center cohort study. COPD patients were enrolled within 72 h of hospitalization. The baseline assessments were Kaolin-TEG and blood samples. Statistical analysis involved using descriptive statistics; the main analysis was a paired *t*-test comparing coagulation parameters between exacerbation and follow-up. One hundred patients participated, 66% of whom were female, with a median age of 78.5 years and comorbidities including atrial fibrillation (18%) and essential arterial hypertension (45%), and sixty-five individuals completed a follow-up after discharge. No significant variations were observed in Kaolin-TEG or conventional coagulation markers between exacerbation and follow-up. The Activated Partial Thromboplastin Clotting Time (APTT) results were near-significant, with *p* = 0.08. In conclusion, TEG parameters displayed no significant alterations between exacerbation and follow-up.

## 1. Introduction

Chronic Obstructive Pulmonary Disease (COPD) is a progressive respiratory disorder characterized by persistent airflow limitation [1]. With a growing global prevalence and presenting a significant healthcare burden, COPD exacerbations are important critical events that lead to increased morbidity, an increase in mortality up to 12% in the first year after a severe exacerbation, and higher healthcare costs [1,2,3,4,5,6]. An exacerbation of COPD may be triggered by respiratory infections, air pollution, or exposure to irritants, further compromising lung function in affected individuals [7]. COPD exacerbations arise from a complex interplay of various pathological processes, including airway inflammation, mucus hypersecretion, and parenchymal destruction [7,8,9,10]. These events ultimately contribute to the impairment of gas exchange and increased airway resistance, leading to a rapid deterioration in respiratory function. However, studies suggests that coagulation abnormalities may also play a significant role in exacerbation pathogenesis [9,10,11,12,13,14,15,16,17,18].

Subsequently, pulmonary embolism (PE), a potentially life-threatening condition, has been observed to occur more frequently during COPD exacerbations in individuals with COPD than in individuals without COPD, with the risk being up to 25% higher in patients with COPD exacerbation [9,10,11,12,13,14,15,16,17,18]. This suggests an intricate connection between thrombotic events and the acute worsening of COPD symptoms resulting in an exacerbation of COPD [9]. Understanding the underlying mechanisms and identifying appropriate diagnostic markers for these coagulation abnormalities are crucial for effective management and improved outcomes.

Coagulation markers, such as D-dimer, fibrinogen, Activated Partial Thromboplastin Time (APTT), thrombocytes, and the International Normalized Ratio (INR) provide quantitative measurements of specific clotting factors and degradation products, reflecting the overall coagulation statuses of patients. However, these conventional coagulation tests have limitations in regard to detecting the dynamic changes and individual variations in clot formation and dissolution.

Thrombelastography (TEG) analysis, on the other hand, is a comprehensive viscoelastic assay that assesses the entire coagulation process, offering a detailed evaluation of clot formation, strength, and lysis by measuring parameters such as the time until the beginning of clot formation (reaction time, R), the time of clot formation (clot formation time, K), clot strength (maximum amplitude), and the lysis of a clot over 30 min (Lys 30), offering insight into the complex interplay between coagulation and fibrinolysis during acute disease [19,20,21]. TEG analysis, therefore, might be able to provide a more accurate representation of the coagulation profile during COPD exacerbations. This functional assessment could enable a better understanding of hypercoagulability and potential fibrinolytic imbalances that contribute to the increased risk of pulmonary embolism in patients presenting with COPD exacerbation.

This study aims to explore the differences in TEG parameters and coagulation markers in COPD patients during admission with exacerbation and at a follow-up after discharge.

## 2. Results

In total, 100 participants were included in this study: 66% of them were female, 38% were current smokers, and the median age was 78.5 years (range 70–82). Atrial fibrillation was detected prior to baseline in 18% of the participants, and 45% had been diagnosed with essential arterial hypertension prior to baseline (Table 1). Approximately 34% received antithrombotic therapy, 22% received anticoagulant therapy, and 12% were administered antiplatelet therapy, while 4% started anticoagulant therapy after enrolment in the study. Table 2 provides the baseline characteristics for the participants receiving antithrombotic therapy at baseline.

### 2.1. Main Analysis

Of the 100 patients initially assessed with a preliminary TEG analysis during admission, 65 individuals were successfully reassessed during follow-up one month after discharge to establish a control reference. Twenty-one participants died before completing the follow-up period, and an additional fifteen declined to participate in the follow-up assessment. There was a mean difference of MA 0.41 (95% CI: −1.42–+2.25, *p* = 0.65). The R mean difference was −1.42 (95% CI: −3.70–+0.86, *p* = 0.22). The results of the paired t-test for all markers are presented in Table 3. The APTTs were numerically different, with a lower APTT for participants during an acute exacerbation, with a mean difference of −4.3 (95% CI: −9.19–+0.62, *p* = 0.08), as shown in Table 3. Figure 1 and Figure 2 illustrate the coagulation parameters. Table 4 shows *p*-values from the paired t-test and Wilcoxon two-sample test.

### 2.2. Sensitivity Analyses

Table 5 demonstrates how the covariates mentioned in the Methodology section affect the coagulation parameters during exacerbation.

## 3. Discussion

In this multi-center, prospective cohort study on patients with COPD, incorporating real-time TEG analysis, we found no differences in clot formation time (R) or clot strength (MA) between exacerbation and follow-up. Additionally, fibrinolysis determined through TEG analysis (Lys 30) exhibited no variance between these phases. The observed increase in clot strength did not seem to be different in either the follow-up stage or the exacerbation stage. Interestingly, APTT was numerically lower in the exacerbation stage, which could signal a slight hypercoagulable state, which fits well with the documented high risk of PE among patients with COPD exacerbation [14,15,17,22]. Although this numeric difference was within the range of clinical meaningfulness, it did not quite meet statistical significance.

Fibrinogen, D-dimer, and INR levels did not show statistically significant differences between the exacerbation and follow-up after discharge. However, in the case of APTT, we observed a near-significant change between exacerbation and follow-up.

Previous research has identified a trend of hypercoagulation in patients with COPD [21,22,23,24]. Our study aligns with the perception of increased fibrin production in COPD patients compared to individuals of similar age without COPD, as the mean D-dimer during exacerbation was 2.2 mg FEU/L and that during follow-up approximately the same, suggesting chronic hypercoagulation since D-dimer levels in individuals of the same age without COPD range from 0.6 to 0.9 mg FEU/L [25]. However, the chronic increase in D-dimer levels was not reflected in APTT, INR, or antithrombin levels. Notably, the mean fibrinogen level ranked in the upper end of normal, which is 12.0 µmol/L. In terms of TEG, a previous study demonstrated a decrease in clot formation time measured through TEG analysis during exacerbation in COPD patients [21]. The referenced study focused on a group hospitalized with an exacerbation (100 individuals) and compared them to a control group of COPD patients in a stable phase (80 individuals) [21]. This setup provided insight into fibrinogenesis and fibrinolysis during an exacerbation; however, it failed to investigate the potential changes in the dynamics of the coagulation cascade from exacerbation to post-discharge in each patient. In our study, which was stronger in its design than the previous study, applying repeated measurements for each participant, the kaolin-TEG parameters revealed no changes in the viscoelastic measurements of the intrinsic cascade at exacerbation and follow-up for the COPD patients. However, we did find a near-significant trend of decreasing APTT, which may hint that despite the neutral findings in the TEG-measurements, the intrinsic cascade may be affected, which is in alignment with earlier research [26,27]. The main difference between our patients at the onset of exacerbation and at follow-up is the treatment in the acute setting, where corticosteroids are suspected to alter the coagulation towards a hypercoagulable state. However, in healthy men, our group recently showed that short-term corticosteroid treatment did not alter kaolin-TEG measurements [28].

To the best of our knowledge, this is the first study observing kaolin-TEG parameters in participants during an exacerbation of COPD, who also served as their own controls. This unique approach enabled the mitigation of numerous confounding factors, thereby directing our attention solely towards the coagulation mechanism. By employing this method, we were able to isolate and evaluate coagulation dynamics with greater precision. Additionally, we have data available on comorbidities and medication that made it possible to investigate the effect of anticoagulation treatment, atrialfibrillation, and previous venous thromboembolism on the study design. This study also has several limitations. Firstly, a considerable number of the participants were unable to attend their scheduled follow-up visits since approximately 20% passed away before the follow-up assessments could be conducted. This circumstance may have introduced attrition bias into this study. It is probable that the participants facing severe health challenges were more inclined not to participate in the follow-up, potentially limiting our understanding of the entire sample size to those with adequate resources for completing the follow-up, which may have introduced a type II error into our study. The near–significant APTT may be a type II error. Secondly, some of the citrate plasma samples used to analyze the conventional coagulation markers were not prepared within the specified time frame, remaining at room temperature for an extended duration beyond the recommended guidelines. Unfortunately, the specific tests affected by this delay remain unidentified. According to Danish guidelines, blood tests that measure anti-thrombin levels, d-dimer levels, and APTT should not be exposed to room temperature for more than four hours, while fibrinogen exposure and the INR should not exceed 24 h. Deviation from guidelines interferes with a precise interpretation of the mentioned coagulation markers; however, we assume that there was a non-different distribution of the deviation. Thirdly, the follow-up time was set to 30 days; however, many COPD patients experience exacerbation symptoms beyond this timeframe. Achieving stability may require an extended duration, particularly for older participants with increased morbidity [29]. It is possible that the intrinsic coagulation pathway may also require an extended duration to stabilize under these circumstances.

The complexities within the coagulation system uncovered in this study emphasize the need for deeper understanding, acknowledging the probable involvement of factors like inflammation, hypoxia, and perhaps even something as simple as immobility in aggravating the intricate interplay observed. Understanding these intricate mechanisms is critical for enhancing COPD management and the development of targeted interventions.

## 4. Materials and Methods

### 4.1. Study Design

This study is a substudy of the ongoing multi-center trial CODEX-P (COPD exacerbation and Pulmonary Hypertension, clinicaltrials.gov: NCT04538976). CODEX-P is an observational multicenter trial that seeks to compare echocardiographic parameters in individuals during COPD exacerbation and at follow-up after discharge. The primary aim is to evaluate changes in echocardiographic measurements between acute exacerbation and follow-up [30]. In the current substudy, the aim was to assess whether functional coagulation was altered during severe COPD exacerbation.

### 4.2. Sample Size

We chose to include the first 100 patients from the CODEX-P trial in this substudy. Power calculation was performed based on the maximal amplitude (MA) in TEG analysis, using paired T-test and the following assumptions: level of significance—5%, power—80%, two-sided statistics, a standard deviation of 4.0 mm, and a detectable difference of 2.5 mm. Based on this, we calculated a required sample size of 82 participants. To account for potential loss to follow-up, we aimed to enroll 100 participants.

### 4.3. Study Participants

A total of 100 consecutive patients were enrolled between January 2020 and October 2023 in a multicenter study initiated at Copenhagen University Hospital Gentofte, with active participation from Copenhagen University Hospital Herlev, Odense University Hospital, and Aarhus University Hospital.

The inclusion criteria were COPD determined through spirometry, confirmed by a Respiratory Physician and in accordance with a recommendation from the Global Initiative for Chronic Obstructive Lung Disease [4] and included within the first 72 h of their hospitalization, and the ability to provide informed consent.

Exclusion criteria were diagnosis with primary pulmonary hypertension using right-heart catheterization [31], moderate to severe heart valve disease, or heart failure prior to enrollment. Men under 40 years old, women under 55 years old, and non-menopausal women over 55 were also excluded from participation.

### 4.4. Study Material

As part of the study protocol, biobank and Kaolin-TEG analyses were conducted during hospitalization (baseline), and a follow-up assessment was scheduled one month following discharge. In cases where patients were unable to visit the hospital for the follow-up assessment, home visits were arranged to facilitate blood sample collection.

Kaolin-TEG results had to be analyzed within two hours after they were obtained from the study participants. TEG samples were citrated (3.2% sodium citrate) whole-blood samples. They were transported to the blood bank at Copenhagen University Hospital Herlev via car, where they underwent immediate analysis. This transportation was executed within a maximum of 15 min after collection, with a total transit duration of 20 min. The entire transportation process was completed within the span of maximum 60 min, ensuring strict adherence to the criteria for analysis within a two-hour timeframe. TEG results were analyzed using the Kaolin TEG assay (TEG^®^ 5000 Hemostasis analyzer, Haemonetics, Boston, MA, USA). Eighteen ROTEM analyses were performed at Odense University Hospital and Aarhus University Hospital; the results of these analyses were further analyzed at the Blood banks of the recruiting hospitals using ***. However, due to low participation in the follow-ups (five individuals participated), the samples were not used in the main analysis.

Fibrinogen, INR, APTT, D-dimer, and antithrombin analyses were performed using frozen 3.2% citrate plasma from the study biobank that was stored in a −80 °C freezer and evaluated in a stable state during long-term freezing [32]. Prior to freezing, biobank samples were centrifuged for 10 min at 2500× *g* and 4 °C. Fibrinogen was measured using the Siemens Dada thrombin assay, INR using the MediRox Owrens PT assay, D-dimer using Siemens INNOVANCE D-dimer assay. APTT was measured using the Siemens Dada Actin FS assay. Antithrombin was measured using the Siemens Standard Human Plasma ORKL, Antithrombin (INNOVANCE Antithrombin). 

### 4.5. Statistical Analysis

Descriptive analysis data was conducted of the demographic and clinical characteristics of the study participants at the date of study entry (baseline). Categorial variables were presented as counts and percentages, while continuous variables were presented as mean and standard deviation if normal distributed. If a normal distribution was not detected, continuous variables were represented using median and quartile range.

#### 4.5.1. Main Analysis

A paired *t*-test was used to assess if coagulation parameters differed between exacerbation and follow-up. The Wilcoxon two-sample test was utilized to assess the coagulation markers, accounting for cases where the assumption of normal distribution was not met. Participants who died or refused to attend the follow-up were excluded from these analyses.

A *p*-value of 0.05 was considered statistically significant.

#### 4.5.2. Sensitivity Analysis

To assess sensitivity, we used analysis of covariance (ANCOVA) to model the coagulation parameters during an exacerbation adjusted for the value at follow-up and the following covariates: age, sex, BMI, smoking status, atrial fibrillation, diabetes mellitus, antithrombotic treatment, essential arterial hypertension, and previous venous thromboembolism.

#### 4.5.3. Data Analysis

All analyses and illustrations were performed using Statistical Analysis Software (SAS) Enterprise 7.1 Guide.

## 5. Conclusions

In conclusion, TEG parameters displayed no significant alterations between exacerbation and follow-up. The nuanced variance in APTT hints at the intricate influence of exacerbation on coagulation dynamics, urging further exploration.

## Figures and Tables

**Figure 1 ijms-25-02051-f001:**
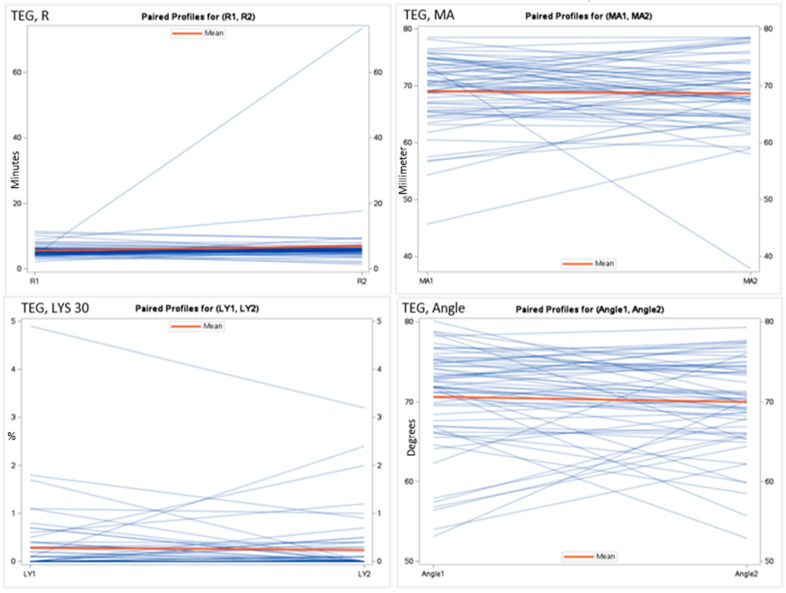
Spaghetti plot for the thrombelastography measurements. Reaction time (R), maximal amplitude (MA), lysis in 30 min (LYS 30), and angle during exacerbation and at follow-up.

**Figure 2 ijms-25-02051-f002:**
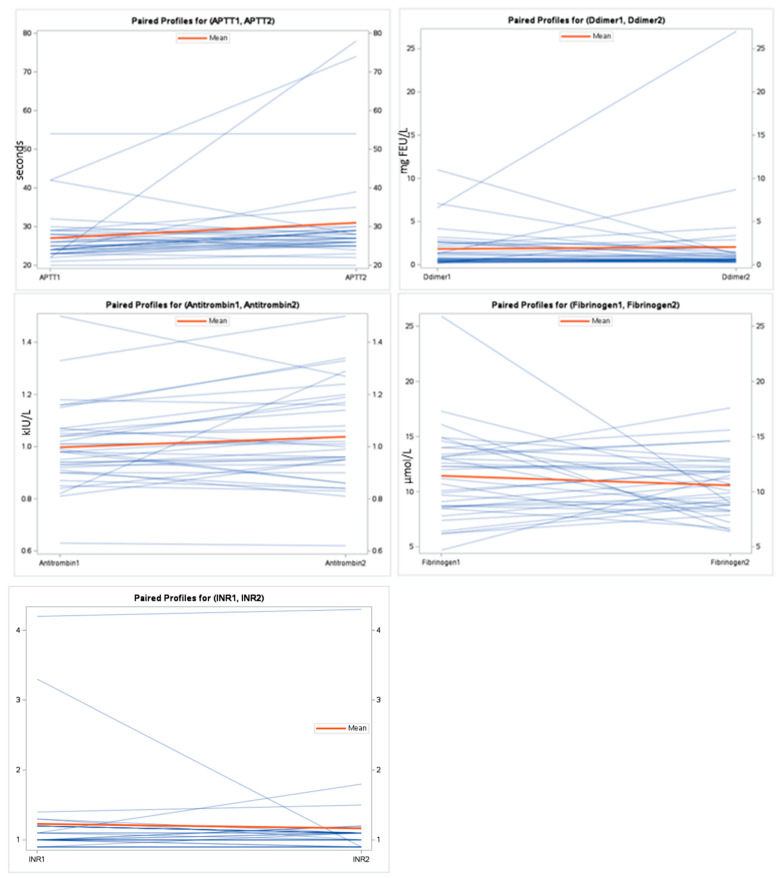
Spaghetti plot for the coagulation markers during exacerbation and at follow-up.

**Table 1 ijms-25-02051-t001:** Demographic characteristics of participants.

	Number of Participants *n =* 100
**Sex, *n* (%)**	
Female	66 (66)
**Age, median (range)**	78.5 (70–82)
≤62, *n* (%)	11 (11)
63–70, *n* (%)	18 (18)
71–77, *n* (%)	24 (24)
≥78, *n* (%)	47 (47)
**BMI, median (range)**	23.7 (20.8–29)
≤18.4 kg/m^2^, *n* (%)	18 (18)
18.5–24.9 kg/m^2^, *n* (%)	45 (45)
25–30 kg/m^2^, *n* (%)	16 (16)
>30 kg/m^2^, *n* (%)	20 (20)
Missing value, *n* (%)	1 (1)
**Smoking status, *n* (%)**	
Former smoker/non-smokers	61 (61)
Current smokers	38 (38)
Missing value	1 (1)
**GOLD stage, *n* (%)**	
GOLD 1	3 (3)
GOLD 2	26 (26)
GOLD 3	47 (47)
GOLD 4	17 (17)
Missing value	7 (7)
**Comorbidities, *n* (%)**	
Diabetes mellitus	13 (13)
Asthma	17 (17)
Atrial fibrillation	18 (18)
Previous VTE	7 (7)
Essential hypertension	45 (45)
Previous cancer	8 (8)
**Medication, *n* (%)**	
Antithrombotic treatment	34 (34)
Anticoagulant therapy	22 (22)
Antiplatelet therapy	12 (12)
Anticoagulant therapy after enrolment *	4 (4)

Abbreviations: VTE, venous thromboembolism. * Four participants started anticoagulant therapy after enrolment.

**Table 2 ijms-25-02051-t002:** Demographic characteristics of participants receiving antithrombotic therapy.

	Number of Participants with Antithrombotic Treatment
	*n = 34*
**Sex, *n* (%)**	
Female	22 (64.7)
**Age, median (range)**	79.5 (72–82)
**BMI, median (range)**	23.4 (21–29)
**Smoking status, *n* (%)**	
Former smoker/non-smoker	21 (61.8)
Current smoker	12 (35.3)
Missing value	1 (2.9)
**GOLD stage, *n* (%)**	
GOLD 1	2 (5.9)
GOLD 2	11 (32.4)
GOLD 3	16 (47.1)
GOLD 4	2 (5.9)
Missing value	3 (8.8)
**Comorbidities, *n* (%)**	
Diabetes mellitus	5 (14.7)
Asthma	7 (20.6)
Atrial fibrillation	18 (52.9)
Previous VTE	7 (20.6)
Essential hypertension	21 (61.8)
Previous cancer	3 (8.8)
**Medication, *n* (%)**	
Anticoagulant therapy	22 (64.7)
Antiplatelet therapy	12 (35.3)

Abbreviations: VTE, venous thromboembolism.

**Table 3 ijms-25-02051-t003:** Difference between coagulation markers during exacerbation and follow-up. Effect estimates and *p*-values. Mean and standard deviation (SD) during an exacerbation and in a stable phase. Mean differences with 95% Confidence Intervals (CIs) and *p*-values.

Marker	Exacerbation, Mean ± SD	Follow-Up, Mean ± SD	Mean Difference (95% CI)	*p*-Value
MA (mm)	69.06 ± 6.2	68.64 ± 6.6	0.41 (−1.42–+2.25)	0.65
R (min)	5.56 ± 1.7	6.98 ± 8.8	−1.42 (−3.70–+0.86)	0.22
Angle (degrees)	70.63 ± 6.3	69.98 ± 5.6	0.64 (−0.92–+2.21)	0.41
Lys 30 (%)	0.28 ± 0.7	0.24 ± 0.6	0.04 (−0.11–0.18)	0.59
Fibrinogen (µmol/L)	11.23 ± 4.2	10.19 ± 2.6	1.05 (−0.76–+2.86)	0.25
Antithrombin (kIU/L)	1.00 ± 0.2	1.02 ± 0.2	−0.02 (−0.07–+0.02)	0.31
D-dimer (mg FEU/L)	2.04 ± 2.5	2.40 ± 5.1	−0.37 (−2.24–+1.51)	0.69
INR	1.16 ± 0.4	1.08 ± 0.2	0.08 (−0.12–+0.27)	0.42
APTT (seconds)	26.68 ± 5.2	30.96 ± 13.2	−4.29 (−9.19–+0.62)	0.08

Abbreviations: R, reaction time; MA, maximum amplitude; Lys 30, clot actively lysed after 30 min; INR, International Normalized Ratio; APTT, Activated Partial Thromboplastin Time.

**Table 4 ijms-25-02051-t004:** *p*-values for coagulation markers during exacerbation and follow-up, showing the results of a paired t-test and a Wilcoxon two-sample test, respectively.

	Paired *t*-Test	Wilcoxon Two-Sample Test
Marker	*p*-Value	*p*-Value
MA (mm)	0.65	0.67
R (min)	0.22	0.11
Lys 30 (%)	0.59	0.08
Angle (degrees)	0.41	0.54
D-dimer (mg FEU/L)	0.69	0.97
Antithrombin (kIU/L)	0.31	0.38
Fibrinogen (µmol/L)	0.25	0.23
APTT (seconds)	0.08	0.12
INR	0.42	0.38

Abbreviations: R, reaction time; MA, maximum amplitude; Lys 30, clot actively lysed after 30 min; INR, International Normalized Ratio; APTT, Activated Partial Thromboplastin Time.

**Table 5 ijms-25-02051-t005:** Analysis of Covariance (ANCOVA) was employed to evaluate the influence of the specified covariates on coagulation markers during exacerbation. The findings are illustrated alongside corresponding *p*-values.

	*p*-Value								
Marker	Age	Sex	BMI	Smoking Status	Atrial Fibrillation	Diabetes Mellitus	Antithrombotic Treatment	Essential Hypertension	Previous VTE
MA (mm)	0.00	0.58	0.46	0.49	0.94	0.60	1.00	0.44	0.22
R (min)	0.64	0.49	0.97	0.92	0.41	0.34	0.21	0.30	0.50
Lys 30 (%)	0.03	0.29	0.18	0.44	0.03	0.11	0.59	0.03	0.54
Angle (degrees)	0.01	0.98	0.95	0.49	0.87	0.28	0.66	0.48	0.23
D-dimer (mg FEU/L)	0.17	0.29	0.56	0.00	0.35	0.14	0.16	0.86	0.60
Antithrombin (kIU/L)	0.33	0.57	0.47	0.62	0.74	0.56	0.43	0.77	0.11
Fibrinogen (µmol/L)	0.11	0.22	0.38	0.57	0.18	0.82	0.93	0.41	0.82
APTT (seconds)	0.32	0.41	0.01	0.63	0.89	0.01	0.10	0.37	0.32
INR	0.63	0.86	0.27	0.85	0.47	0.26	0.14	0.13	0.80

Abbreviations: R, reaction time; MA, maximum amplitude; Lys 30, clot actively lysed after 30 min; INR, International Normalized Ratio; APTT, Activated Partial Thromboplastin Time; VTE, venous thromboembolism; BMI, Body Mass Index.

## Data Availability

We believe that knowledge sharing increases the quality and quantity of scientific results. The sharing of relevant data will be discussed within the study group upon reasonable request.

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
