# Peer review of "Thrombelastography and Conventional Coagulation Markers in Chronic Obstructive Pulmonary Disease: A Prospective Paired-Measurements Study Comparing Exacerbation and Stable Phases"

_ijms, 2024, doi:10.3390/ijms25042051_

Round 1
Reviewer 1 Report
Comments and Suggestions for Authors
In this manuscript, the authors performed group paired studies of kaolin-TEG and coagulation marker analysis in COPD between the time of exacerbation and stable time. The study is well designed. Although the authors have not found significant differences in terms of coagulation index, it is still valuable to the COPD community.
I only have minor suggestions regarding the figures.
1. Two figures (figure 2a, figure 2b) should be labeled as figures 1 and 2.
2. The labels for the Y axis should be included in each graph to clearly show the meaning of the values in the Y axis.
3. Current graphs only show samples with follow-up data. It would also be valuable to include all data points in the graph to compare initial measurements between patients who attended follow-up and those who did not.
Author Response
Thank you for recognising the importance of the work. I have replied to the comments as follows:
Two figures (figure 2a, figure 2b) should be labeled as figures 1 and 2.
Thank you for the comments. The labels of the figures have now been changed, please see page 7 and 8 in the article, where this is written.
The labels for the Y axis should be included in each graph to clearly show the meaning of the values in the Y axis.
Thank you for this comment, it has now been added to figure 1 and 2.
Current graphs only show samples with follow-up data. It would also be valuable to include all data points in the graph to compare initial measurements between patients who attended follow-up and those who did not.
This is a very good point, thank you. However, the focus of this article was to compare the during exacerbation with the follow-up phase. For this reason, we did not make graphs with only initial measurements.
Reviewer 2 Report
Comments and Suggestions for Authors
This is a well writen manuscript with pretty robust sections overall. Also, the author discussed the outcome of TEG analysis in patient cohort before and after follow up.
The method used in the manuscript is important to understand the risk of hypertension during the COPD exacerbation. however, the results show no difference among all different conditions and groups. Although the author discussed the results and provided a strong background support, it is how to know if there is any correlation of pulmonary embolism with COPD exacerbation. The difference can not be detected by TGE or there is actually no difference at all.
The conclusion is supported by the results, but a bit more positive results could be more helpful to understand the correlation of the diseases authors pointed in this manuscript.
Author Response
Thank you very much for this comment and this reflection. We agree that a positive sign would have been a great addition to the knowledge of COPD during an exacerbation. Our study did however, show an interesting insight in the coagulation system, and as it is one of the few articles on this topic we find it very relevant and interesting.
Round 2
Reviewer 2 Report
Comments and Suggestions for Authors
Thanks for addressing the responses. However, I still think some positive results would be very helpful, which can be extracted from the negative results. Please consider it.
Author Response
Thank you very much for this comment, and for wanting to improve our manuscript. We have investigated it, and the following has been added:
- The reporting of results in the result section has been optimized by adding the actual numbers of our findings and not our interpretation of the numbers (se line 97-103).
- In the discussion is has been clarified that the loss to follow-up can have resulted in a type II error, leading to APTT with a p-value 0.08 (see line 238-239).
- In the discussion the lines 192-196 have been added to clarify that there is a numeric increase in APTT at follow-up
- We have added a table 4 showing the p-value of the paired t-test and the Wilcoxon test, respectively (between line 134-137).
- We have added table 5 to illustrate the p-values from the ANCOVA (line 144-157).
Round 3
Reviewer 2 Report
Comments and Suggestions for Authors
Thanks for addressing my concerns and it looks good now!